# AI Enhances Lung Ultrasound Interpretation Across Clinicians with Varying Expertise Levels

**DOI:** 10.3390/diagnostics15172145

**Published:** 2025-08-25

**Authors:** Seyed Ehsan Seyed Bolouri, Masood Dehghan, Mahdiar Nekoui, Brian Buchanan, Jacob L. Jaremko, Dornoosh Zonoobi, Arun Nagdev, Jeevesh Kapur

**Affiliations:** 1Exo Imaging, Santa Clara, CA 95054, USA; 2Department of Critical Care Medicine, Faculty of Medicine and Dentistry, University of Alberta, Edmonton, AB T6G 2B7, Canada; 3Department of Radiology and Diagnostic Imaging, Faculty of Medicine and Dentistry, University of Alberta, Edmonton, AB T6G 2R3, Canada; 4Alameda Health System, Highland Hospital, University of California San Francisco, San Francisco, CA 94143, USA; 5Department of Diagnostic Imaging, National University of Singapore, Singapore 119074, Singapore

**Keywords:** lung ultrasound, artificial intelligence, pleural effusion, consolidation, diagnostic accuracy

## Abstract

**Background/Objective**: Lung ultrasound (LUS) is a valuable tool for detecting pulmonary conditions, but its accuracy depends on user expertise. This study evaluated whether an artificial intelligence (AI) tool could improve clinician performance in detecting pleural effusion and consolidation/atelectasis on LUS scans. **Methods**: In this multi-reader, multi-case study, 14 clinicians of varying experience reviewed 374 retrospectively selected LUS scans (cine clips from the PLAPS point, obtained using three different probes) from 359 patients across six centers in the U.S. and Canada. In phase one, readers scored the likelihood (0–100) of pleural effusion and consolidation/atelectasis without AI. After a 4-week washout, they re-evaluated all scans with AI-generated bounding boxes. Performance metrics included area under the curve (AUC), sensitivity, specificity, and Fleiss’ Kappa. Subgroup analyses examined effects by reader experience. **Results**: For pleural effusion, AUC improved from 0.917 to 0.960, sensitivity from 77.3% to 89.1%, and specificity from 91.7% to 92.9%. Fleiss’ Kappa increased from 0.612 to 0.774. For consolidation/atelectasis, AUC rose from 0.870 to 0.941, sensitivity from 70.7% to 89.2%, and specificity from 85.8% to 89.5%. Kappa improved from 0.427 to 0.756. **Conclusions**: AI assistance enhanced clinician detection of pleural effusion and consolidation/atelectasis in LUS scans, particularly benefiting less experienced users.

## 1. Introduction

Over the past decade, lung ultrasound (LUS) has become a valuable bedside tool, particularly in emergency departments (EDs) and intensive care units (ICUs) [1,2]. It offers a radiation-free, low-cost, rapid, and portable method for real-time assessment of pulmonary structures [3,4,5]. Compared to chest radiography, LUS demonstrates higher sensitivity and similar specificity for detecting conditions such as pleural effusion, pneumonia, and pulmonary edema [6,7,8,9,10]. The Bedside Lung Ultrasound in Emergency (BLUE) protocol [11] provides a standardized LUS approach in critical care, delivering over 90% diagnostic accuracy for pneumonia, pneumothorax, pulmonary edema, and pulmonary embolism, all in under three minutes [12].

Ultrasound is particularly effective in visualizing pleural fluid. Simple effusions appear anechoic, while complex effusions (e.g., chronic, malignant, hemithorax, empyema) often present with heterogeneous features [13,14]. LUS is superior to chest radiography in detecting and evaluating the type of pleural effusions [7], aiding in interventions like drainage or more aggressive treatments [4,7].

Lung consolidation and sometimes gross atelectasis can cause the lung to appear as solid, non-aerated tissue on ultrasound, resembling the liver, a phenomenon known as hepatization [4]. In these conditions, the normally air-filled lung becomes visible as it becomes a dense and hypoechoic structure [4]. Lung consolidation can result from diseases such as pneumonia, infarction, or tumor infiltration. Atelectasis, which involves the collapse or loss of air in parts or all of the lungs, can be caused by compressive forces (e.g., a large pleural effusion) or internal airway blockages [15].

As LUS use expands, many clinicians advocate for its routine application in ICUs to reduce reliance on chest radiographs, which carry increased costs and cumulative radiation exposure, particularly in pediatric populations [5,16]. However, LUS image quality and diagnostic accuracy remain highly dependent on user expertise. AI has emerged as a potential solution, with tools capable of guiding scan acquisition and detecting conditions such as pleural effusion and consolidation in real time [17,18,19,20,21]. Real-time AI assistance in interpreting LUS scans for conditions like pleural effusion and consolidation/atelectasis could improve accuracy, reliability, and efficiency, regardless of the user’s expertise.

Previous studies have demonstrated the potential of deep learning (DL) models in diagnosing conditions specific to lung ultrasound (LUS). Arntfield et al. developed a DL model that could distinguish between A-line and B-line artifacts [22]. Ebadi et al. proposed a multiclass DL model capable of classifying A-line, B-line, and pleural effusion or consolidation [23]. Shang et al. showed that such models could improve radiologists’ accuracy in identifying COVID-19 pneumonia on ultrasound images [24]. Chaudhary et al. introduced Pleff-Net, a DL model specifically designed to detect pleural effusions of varying sizes [25]. Hong et al. developed a model that improved radiologists’ accuracy for the detection of pleural effusion and consolidation [17]; however, their evaluation was limited to only two radiologists, raising questions about generalizability. More comprehensive reviews of state-of-the-art AI applications in LUS are available in recent survey papers [26,27].

Despite promising results, a key limitation across the existing literature is that most models are evaluated as stand-alone systems, without integration into clinical workflows or real-time interaction with human users. No prior work has employed a multi-reader, multi-case (MRMC) design to systematically assess how AI tools influence diagnostic performance across clinicians with different levels of ultrasound expertise.

This represents an important gap that remains underexplored, especially since LUS is routinely performed in real-world settings by users with varying levels of expertise. To address this gap, the present study investigates whether an AI-based computer-aided detection (CADe) system can enhance diagnostic accuracy for pleural effusion, consolidation, or atelectasis in LUS, specifically through an MRMC study involving 14 readers with varying levels of training in LUS from novice to experts.

## 2. Materials and Methods

### 2.1. Study Design

We retrospectively collected 465 LUS scans from 359 patients, aged 18 to 75, across six United States and Canadian centers between January 2019 and July 2024. Lung scans from these external centers were entirely segregated from the data used for internal training/tuning of the AI algorithm. Cases were selected consecutively in the order they were received. Following the BLUE protocol, only PLAPS (Posterolateral Alveolar and/or Pleural Syndrome) point images, known to be sensitive to detecting pleural effusion and consolidation, were included. The dataset was then enriched to include at least 30% abnormal cases per center to ensure sufficient representation of clinically relevant conditions. Scans were acquired using curvilinear, phased-array, or portable transducers (1.5–7 MHz) with Exo Iris (86 scans), Sonosite X-Porte (253 scans), and Philips Lumify (126 scans). Scans that did not meet the quality standards of the American College of Emergency Physicians (ACEP) or the American Institute of Ultrasound in Medicine (AIUM) were excluded. For the MRMC study, a random subset of 374 cases was selected, consisting of 185 positive and 189 negative cases for pleural effusion, and 185 positive and 189 negative cases for consolidation or atelectasis.

The 14 readers selected for this MRMC study represented a range of expertise, from specialists (trained emergency physicians with fellowship training in LUS) to less experienced users (medical students who received targeted training to identify the relevant pathologies for this study). This diversity was intentionally chosen to reflect the spectrum of potential users and to assess the impact of AI assistance more effectively.

### 2.2. Selection of Participants

The 14 readers participated in two separate reading sessions: one without AI assistance and one with AI assistance, separated by a four-week washout period. In each session, readers assigned a score from 0 to 100 to indicate the likelihood of the condition (either pleural effusion or consolidation/atelectasis) being present in the scan. The first session was conducted without assistance, while the second session incorporated AI-generated bounding boxes provided by the Exo Lung AI (iris 2.5).

For subgroup analysis, readers were divided into three categories based on expertise: novices, experienced, and experts. Novices were medical students with some LUS exposure (R13, R14). Experienced users included residents and physicians who completed extensive didactic and hands-on LUS training during residency and actively use LUS in practice (R2, R5, R6, R7, R8, R11, R12), exceeding ACEP educational standards. Experts were physicians with LUS fellowship training or board certification and years of experience (R1, R3, R4, R9, R10).

### 2.3. Inclusion/Exclusion Criteria

Inclusion: Adult patients aged above 18 years underwent lung ultrasound at one of six participating centers. Scans were included if they captured the PLAPS point in accordance with the BLUE protocol. Cases were collected consecutively as they became available and were later enriched to ensure at least 30% abnormal cases per site, improving representation of diagnostic conditions relevant to the study aims.

Exclusion criteria were limited to scan quality: images were excluded if they failed to meet quality benchmarks defined by ACEP or AIUM; typically, this would be images with large rib shadows or inappropriate depth (too shallow or too deep) that result in poor visibility of the lung or diaphragm.

The final study dataset used for the MRMC analysis was a randomized selection of 374 cases from the eligible pool, with balanced numbers of positive and negative findings for each target condition (pleural effusion, consolidation/atelectasis).

### 2.4. Exo Lung AI Software

The system uses a U-Net style architecture [28] for multi-class segmentation to identify pleural effusion and consolidation/atelectasis across individual frames. In parallel, a landmark detection model identifies key anatomical structures of the lung, such as the diaphragm, liver/spleen, and kidney. Per-frame outputs are then stored and processed collectively at the clip level. During this post-processing phase, a contour validation algorithm assesses the validity of the predicted effusion and consolidation masks based on their relative position to the anatomical structures. A clip is classified as positive for a condition if at least one frame contains a validated contour for the corresponding pathology. The algorithm selects the frame with the largest valid pathological contour for the region of interest placement. The box of the region of interest is then overlaid on the corresponding image and shown to readers during interpretation. An example of these bounding boxes is shown in Figure 1.

### 2.5. Ground Truthing

Three board-certified experts, either radiologists or emergency physicians with a minimum of 10 years of experience and more than 5000 prior LUS reviews, established the ground truth labels for each ultrasound image based solely on the LUS data, without access to any additional information such as CT scans or X-rays. Two experts independently analyzed the scans, assigning one binary label for pleural effusion and another for consolidation or atelectasis (1 indicating presence, 0 indicating absence). In cases of disagreement, a third expert served as a tiebreaker to determine the final label.

### 2.6. Outcomes

The primary objective is to assess Exo Lung AI’s ability to assist clinicians and healthcare professionals in interpreting LUS scans for detecting pleural effusion and consolidation/atelectasis. Subgroup analysis by expertise level was a secondary outcome. Agreement among clinicians and healthcare professionals was also measured before and after AI assistance.

### 2.7. Analysis

Statistical analysis was performed using iMRMC software [29] to estimate mean AUC differences, *p*-values, and confidence intervals within an MRMC framework. A two-sided *p*-value < 0.02 was used to reject the null hypothesis of no AUC difference. The same criteria applied to the experience-based subgroup analysis. A mean AUC difference > 0.02 indicated successful performance. Sensitivity and specificity differences were analyzed using one-sided permutation tests in Python’s SciPy library [30], testing the null hypothesis that the mean difference (aided minus unaided) was less than or equal to zero, based on 100,000 permutations. *p*-values < 0.02 were considered significant. Confidence intervals were computed in R using the DescTools package [31] with 100,000 BCa bootstrap resamples. Fleiss’ Kappa, calculated using Python’s Statsmodels library [32], assessed inter-rater reliability; improvement and values > 0.6 indicated successful agreement.

## 3. Results

### 3.1. Characteristics of Study Subjects

465 LUS scans were collected from 359 patients aged 18 to 75 years (mean age 38.8 ± 14.5 years; 15 percent female, 17 percent male, remainder unspecified) across six United States and Canadian centers between January 2019 and July 2024. All scans were from the PLAPS point, selected consecutively, and following the BLUE protocol [11].

### 3.2. Main Results

The Exo Lung AI algorithm showed high accuracy compared to the ground truth from three experts. For pleural effusion detection, sensitivity was 0.967 (95% CI: 0.94–0.99) and specificity was 0.912 (95% CI: 0.87–0.95). For consolidation/atelectasis detection, sensitivity was 0.968 (95% CI: 0.94–0.99) and specificity was 0.945 (95% CI: 0.91–0.98). The analysis was conducted on a per-image basis. Subgroup analyses confirmed consistent performance across centers, devices, and patient sex, with sensitivity and specificity remaining above 0.94 and 0.83 in all subgroups (Table 1).

The study compared human readers’ performance in detecting pleural effusion with and without AI assistance. The AUC-ROC improved from 0.917 to 0.960 (ΔAUC = 0.044; *p* < 0.01; 95% CI: 0.0142–0.0729), indicating a statistically significant gain (Figure 2A). Sensitivity increased from 77.3% to 89.1% (ΔSe = 11.8%; *p* < 0.01; 95% CI: [5.9%, 18.6%]), and specificity slightly improved from 91.7% to 92.9% (ΔSp = 1.1%; *p* < 0.31; 95% CI: [−1.9%, 4.5%]). AUC-ROC improved in 13 of 14 readers, with 9 showing >2% improvement (Figure 3A). Fleiss Kappa rose from 0.612 to 0.774, reflecting better inter-reader agreement with AI aid. It should be noted that Fleiss Kappa was calculated by converting the readers’ scores into binary labels (positive/negative) using a threshold of 50.

The study compared human readers’ performance in detecting consolidation/atelectasis with and without AI assistance. The AUC-ROC improved from 0.87 to 0.941 (ΔAUC = 0.071; *p* < 0.01; 95% CI: 0.0221–0.1200), showing a statistically significant gain (Figure 2B). Sensitivity increased from 70.7% to 89.2% (ΔSe = 18.4%; *p* < 0.001; 95% CI: [10.6%, 27.5%]), and specificity improved from 85.8% to 89.5% (ΔSp = 3.7%; *p* < 0.11; 95% CI: [−0.005%, 7.5%]). AUC-ROC improved in 11 of 14 readers, with 8 showing >2% improvement (Figure 3B). Inter-reader agreement, measured using Fleiss Kappa, rose from 0.427 to 0.756 after AI assistance.

In MRMC subgroup analyses at a 95% significance level (ɑ = 0.05), AUC improved significantly in pleural effusion detection by 10.8% for novices (*p* < 0.001) and 5% for experienced readers (*p* < 0.02), with all subgroups reaching about 0.96 after AI-assisted reading (Figure 4A). Similarly, in consolidation/atelectasis detection, AUC improved significantly by 20.8% for novices and 7.6% for experienced readers (both *p* < 0.01), with all subgroups reaching about 0.94 post-AI assistance (Figure 4B).

The radar chart comparing unaided and AI-aided sensitivity and specificity for detecting pleural effusion and consolidation/atelectasis per reader is presented in Figure 5.

## 4. Discussion

In this study, human readers of varying experience showed substantially improved performance in detecting pleural effusion and consolidation/atelectasis on previously obtained LUS images when they were provided with AI assistance. In our MRMC analysis, the AI assistance significantly improves sensitivity, particularly for less experienced readers, while slightly improving specificity.

AUC-ROC analysis showed significant improvements in detecting pleural effusion (4.4%) and consolidation/atelectasis (7.1%). Both novice and experienced users benefited, with performance approaching expert level. Exo Lung AI significantly increased average sensitivity, with gains of over 11.8% for pleural effusion and 18.4% for consolidation/atelectasis. Specificity improved modestly, as baseline levels were already high (over 90% for pleural effusion and 85% for consolidation). With AI, both sensitivity and specificity reached approximately 90%. MRMC analysis highlighted greater gains in sensitivity, which is crucial for accurately identifying true cases in clinical care.

There is a body of work on AI systems for interpreting lung images, including LUS or chest radiography [33,34,35,36,37,38]. To the best of our knowledge, our study stands as the first of its kind in terms of the data modality and study design. Several AI systems have been developed to classify ultrasound images by detecting conditions such as pleural effusion or consolidation [17,39,40]. For example, one validated deep learning (DL) model distinguished normal (A-line) from abnormal (B-line) lung patterns on LUS with 93% sensitivity and 96% specificity across 400 clips [41]. Another model diagnosed absent lung sliding on 241 LUS scans, achieving 92.1% sensitivity, 80.2% specificity, and 88.5% accuracy [42]. In a study with a similar design, the impact of an AI assistant on users with diverse backgrounds in identifying different abnormal findings from chest radiographs was examined [36]. In the study described [36], 12 readers reviewed half of the radiographs with AI assistance and half without. AI raised sensitivity by 8.5% and 14%, and specificity by 3.9% and 2.9%, for pleural effusion and consolidation, respectively. In another study [38], an experience-based subgroup analysis showed unaided accuracy ranged from 68.7% (beginners) to 95% (experts) for key LUS lung findings, including pleural effusion and consolidation. They also demonstrated that the accuracy of 57 non-expert users in identifying similar lung findings in LUS clips increased from 68.9% to 82.9% after using their AI-based software [38]. The body of work presented just prior, including both the design elements and general findings, provides a foundation and context that guide the study design outlined in this paper. These findings are also broadly in line with the existing literature, further supporting the relevance of the methodological choices made in this work.

These results indicate that the integration of Exo Lung AI assistance not only enhances the detection capabilities of clinicians but also contributes to more reliable and accurate identification of lung conditions, thereby improving overall diagnostic performance.

### Strengths and Limitations

The study included 14 clinical users with varying training and experience, reflecting the typical range in point-of-care settings. A strength is the use of data from multiple ultrasound device manufacturers.

This study also has limitations. One limitation is that the analysis was performed per clip rather than per patient, as the tool is designed to flag individual clips needing closer attention. Including data from multiple lung regions could enhance detection accuracy and comprehensiveness.

While multi-center data was used, it is important to note that data is limited to a North American population. Including populations from other continents would be of value in the future. Thirdly, no external reference standard (such as X-ray or CT scan) was used. For future studies, we will incorporate cross-validation using gold-standard modalities such as chest X-ray or CT to enhance diagnostic certainty. AI can also support other aspects of the workflow, such as flagging poor-quality images [20] (done manually in this study), providing segmentation masks for conditions and organs, guiding image acquisition, or enhancing information during diagnosis [43]. While the proposed AI algorithm improved diagnostic performance, its outputs should be interpreted as decision support. AI is not there to replace clinical judgment, so we can avoid potential over-reliance or misinterpretation.

Finally, the algorithm was evaluated solely on clips obtained from the PLAPS point. While clinically relevant, this can limit generalization for full lung assessments. For future iterations of the study, we aim to conduct multi-view validation.

## 5. Conclusions

This study found that AI algorithms for the detection of pleural effusion and consolidation/atelectasis in LUS scans enhanced the performance of both novice users and highly trained healthcare providers, particularly by improving sensitivity for these pathologies. Since most clinicians have minimal training in interpreting LUS, our model can be instrumental in upskilling clinicians at the point-of-care and facilitating easier integration of this solution into clinical care. The deployment of such systems alongside existing Point-of-Care Ultrasound (POCUS) solutions can improve patients’ clinical care in current healthcare settings, reduce the need for ionizing radiation, and eventually lower healthcare costs.

## Figures and Tables

**Figure 1 diagnostics-15-02145-f001:**
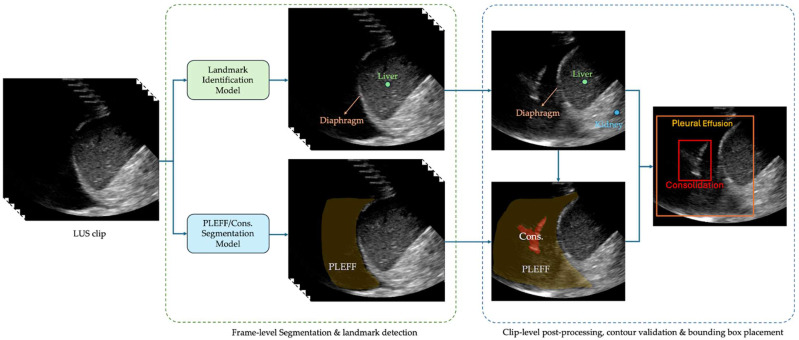
An example of the produced bounding boxes for pleural effusion and consolidation/atelectasis.

**Figure 2 diagnostics-15-02145-f002:**
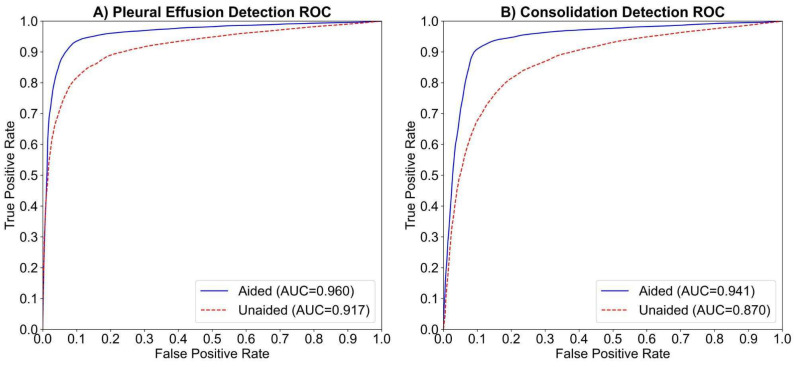
The average ROC curves and AUCs for the aided and unaided readings of 14 readers in detecting (**A**) pleural effusion and (**B**) consolidation/atelectasis, respectively. The dashed red line represents the average ROC curve for unaided readings, while the solid blue line represents the average ROC curve for aided readings.

**Figure 3 diagnostics-15-02145-f003:**
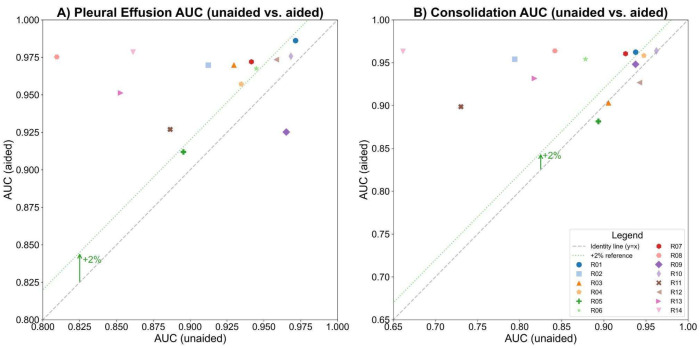
The AUCs before and after AI assistance for each reader in detecting (**A**) pleural effusion and (**B**) consolidation/atelectasis, respectively, with the gray dashed line indicating no change and the green dotted line marking a 2% AUC improvement threshold.

**Figure 4 diagnostics-15-02145-f004:**
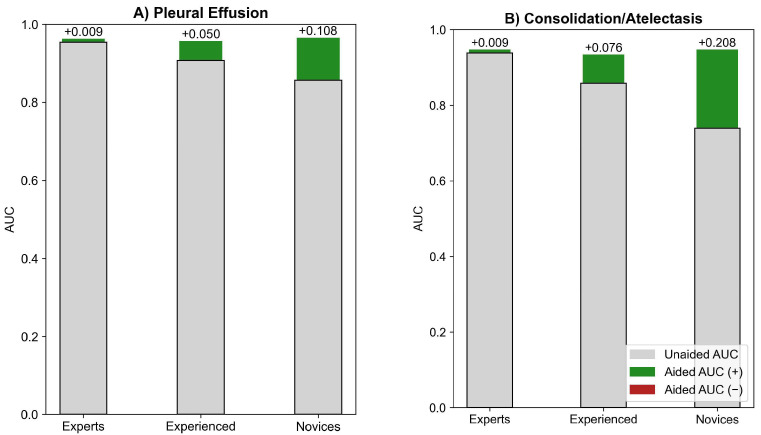
MRMC experience-based subgroup analysis for (**A**) pleural effusion and (**B**) consolidation/atelectasis detection.

**Figure 5 diagnostics-15-02145-f005:**
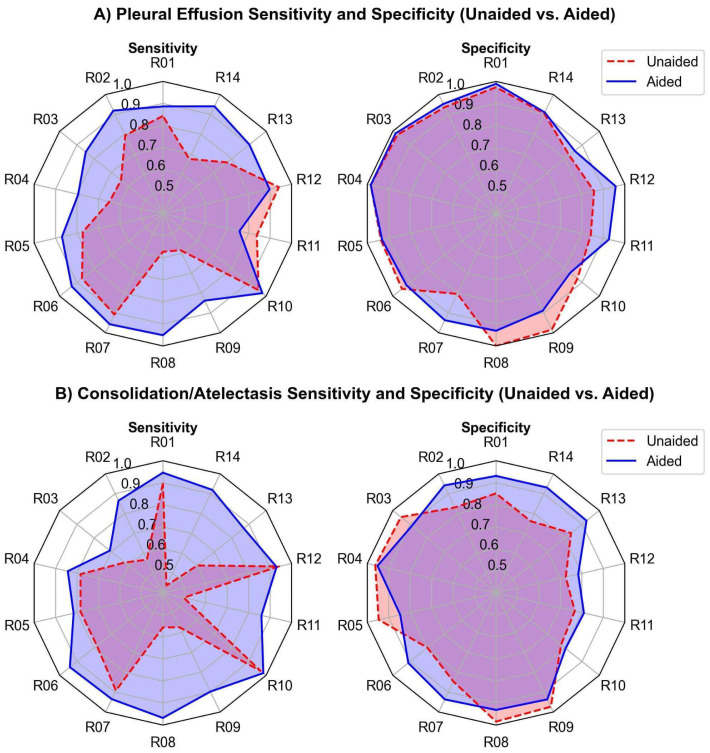
Sensitivity and specificity (unaided vs. aided) for all 14 readers in detecting (**A**) pleural effusion and (**B**) consolidation/atelectasis. The dashed red and solid blue lines represent the unaided and aided reading results, respectively.

**Table 1 diagnostics-15-02145-t001:** Subgroup analyses of the AI results compared to the ground truth.

		Data Source Subgroup (% Data)		Device Manufacturer Subgroup (% Data)			Male/Female Subgroup (% Data)	
		The U.S. (52%)	Canada (48%)	Exo (19%)	Sonosite (54%)	Philips (27%)	Male (17%)	Female (15%)
**Pleural Effusion**	**Specificity** **(95% CI)**	0.94 (0.90–0.98)	0.83 (0.73–0.93)	0.93 (0.85–0.99)	0.85 (0.76–0.93)	0.95 (0.90–0.99)	0.92 (0.84–0.99)	0.92 (0.80–0.99)
	**Sensitivity** **(95% CI)**	0.97 (0.93–0.99)	0.96 (0.93–0.99)	0.97 (0.88–0.99)	0.97 (0.94–0.99)	0.96 (0.86–0.99)	0.97 (0.88–0.99)	0.98 (0.91–0.99)
**Consolid./Atelec.**	**Specificity** **(95% CI)**	0.97 (0.94–0.99)	0.88 (0.79–0.97)	0.93 (0.85–0.99)	0.90 (0.82–0.98)	0.99 (0.96–0.99)	0.96 (0.89–0.99)	0.92 (0.79–0.99)
	**Sensitivity** **(95% CI)**	0.95 (0.90–0.99)	0.98 (0.95–0.99)	0.94 (0.83–0.99)	0.98 (0.95–0.99)	0.94 (0.83–0.99)	0.94 (0.84–0.99)	0.95 (0.88–0.99)

## Data Availability

The dataset presented in this paper is not publicly available due to privacy concerns and the restrictions outlined in the data sharing agreement the authors have with the medical institution that obtained the data.

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
