# Peer review of "AI Enhances Lung Ultrasound Interpretation Across Clinicians with Varying Expertise Levels"

_diagnostics, 2025, doi:10.3390/diagnostics15172145_

Round 1

Reviewer 1 Report

Comments and Suggestions for Authors

This study evaluates the performance of human readers in detecting pleural effusion and consolidation/atelectasis using AI support to interpret ultrasound data. This research, involving 14 clinical users, considers different levels of training and experience.
My suggestions for this study: 
1-While the article contains significant analyses, the main motivation needs to be presented more comprehensively. The literature review section should be enriched with both current studies in the field and similar artificial intelligence applications.
2-The artificial intelligence method used should be explained in more detail. Although this is a specialized area, detailing the artificial intelligence methods and methodology used in the system will facilitate the analysis of the model's outputs. In particular, section "2.4. Exo Lung AI Software" should be expanded. 

3-Visual outputs generated by the model could be added to the experimental results section. Such visuals will help the reader better understand how the model produces results. 

4-In the discussion section, the strengths and weaknesses of the artificial intelligence model used should be discussed in detail. Evaluating the effectiveness of the model is important for the reliability of the results. Criticisms and suggestions regarding the model should definitely be included.

Reviewer 2 Report

Comments and Suggestions for Authors

1. How generalizable are the results across diverse patient populations and healthcare settings beyond the six centers studied?
2. What are the limitations of AI assistance in LUS interpretation, and are there cases where AI may lead to diagnostic over-reliance or errors?
3. What are the ethical implications of AI-assisted diagnosis in clinical decision-making — particularly in balancing human oversight and AI recommendations?

Reviewer 3 Report

Comments and Suggestions for Authors

Dear Authors,

I have completed my evaluation of your work. I have indicated below the areas that I see necessary in the study. I hope that making the improvements specified in these items will contribute to your work. I wish you success.

Evaluation

• No literature review is presented in the introduction or as a separate section. The sources provided in the introduction do not explain clinical studies parallel to the research or AI-supported clinical studies. Therefore, the study should be expanded by conducting a literature review parallel to the study conducted in this study and presenting it as a separate section or in the introduction.

• The “goal of this investigation” subsection in the introduction section is not numbered. Therefore, if it is not a subheading, it should not be presented in this way. Moreover, there is no need to mention it in this way. The purpose of the study is already stated in the last paragraph.

• It would be useful to mention the contributions of the study to the field in the introduction section.

• Was ethical approval obtained for the study? If so, this should be stated. If not, how were ethical issues addressed?

• The study mentions Exo Lung AI software and states that this software achieves high success rates based on grand truth. The parametric settings of this software are not specified.

• It is stated that the mentioned software support has a positive contribution to all metrics. It is also stated in the study that this software already performs measurements with high accuracy on its own. In this case, can it not be said that the findings obtained are already detected by the software rather than by humans?

• The concept of AI assistance is not entirely clear. What kind of assistance did the artificial intelligence software provide to the readers? Doesn't showing it with a bounding box already mean that this software detected it?

• In Figure 5, differences were observed in the metrics for both shapes in supported and unsupported conditions. In some cases, a negative effect was even observed. The reasons for this have not been sufficiently explained.

• In the discussion section, if there are any, similar AI-assisted diagnosis studies should be compared. The pros and cons of other studies should be discussed. If this study is the first of its kind, this should be mentioned.

Reviewer 4 Report

Comments and Suggestions for Authors

The study presents a well-structured investigation into how AI can improve lung ultrasound (LUS) interpretation for clinicians of varying expertise. However, there are several critical areas that require improvement and elaboration to enhance the rigor and generalizability of the manuscript.

The methodology section, although clear in describing the study setup and reader participation, lacks sufficient detail regarding the internal mechanics of the AI algorithm. Specifically, the architecture of the neural network used in the Exo Lung AI tool remains unspecified. Readers would benefit from a brief but meaningful explanation of the deep learning model such as whether it uses U-Net, ResNet, or a custom CNN and how the bounding boxes were generated. Further, there is no mention of the loss functions, number of training epochs, training dataset volume, or any regularization or augmentation strategies used. Including these would significantly increase transparency and reproducibility.

The reliance on only the PLAPS point for scan analysis, while clinically relevant, limits the broader applicability of the AI tool. Lung ultrasound in routine clinical settings often incorporates multi-view assessments (e.g., anterior, lateral, apical views), and the exclusion of these could hinder generalization. The authors should acknowledge this in the discussion and propose plans for future studies involving multi-view validation.

Moreover, while ground truth was established via experienced LUS reviewers, it was done solely using ultrasound data without cross-reference to gold-standard imaging like CT or chest X-ray. This introduces uncertainty in diagnostic labeling. The authors should address this limitation explicitly, and ideally, consider cross-validation with radiological data in future iterations of this research.

Statistical analysis, while robust, employs a binary threshold of 50 to calculate Fleiss’ Kappa from continuous reader scores. This conversion may oversimplify inter-reader agreement and affect interpretability. The authors should justify this approach and consider exploring more granular agreement measures like weighted or ordinal Kappa in future work.

Another limitation relates to population diversity. The dataset is derived entirely from North American centers, which may restrict the AI model's generalizability to other regions with different scanning practices, devices, or clinical populations. This should be acknowledged in the discussion, along with a statement about extending the dataset to include data from Asia, Africa, and Europe in future studies.

In terms of model trust and explainability, the manuscript would benefit from a clearer discussion of how bounding box outputs align with expert-verified features and how clinicians perceived these outputs during the reader study. Readers would also appreciate knowing whether techniques like Grad-CAM or attention visualization were used to validate model focus during prediction.

Figures 3 and 5 effectively convey performance improvements, but they lack confidence intervals or variance measures for each reader. Including these would help clarify the consistency and statistical strength of the results. At a minimum, the authors should mention this omission and potentially provide reader-specific performance metrics in the supplementary material

Round 2

Reviewer 1 Report

Comments and Suggestions for Authors

The author has addressed all my concerns and made the necessary revisions. I recommend that the manuscript be accepted in its current form.

Reviewer 3 Report

Comments and Suggestions for Authors

Thank you for your efforts.

Reviewer 4 Report

Comments and Suggestions for Authors

Accepted